# Understanding Spatial Characteristics of Refugee Accommodations Associated with Refugee Children’s Physical Activity in Microenvironments: Six Case Studies in Berlin

**DOI:** 10.3390/ijerph19137756

**Published:** 2022-06-24

**Authors:** Siqi Chen, Martin Knöll

**Affiliations:** Urban Design and Planning (UDP), Department of Architecture, Technical University of Darmstadt, El-Lissitzky-Str. 1, 64287 Darmstadt, Germany

**Keywords:** migrants, refugee facilities, space syntax, active play, barriers, design, microenvironment, living environment

## Abstract

Refugee children often spend a considerable amount of time in refugee accommodations with limited space and limited access to communal facilities. Such environmental settings make it difficult for refugee children to engage in physical activity (PA), which is essential for their health and social inclusion. While there is a strong evidence base for environmental attributes associated with non-refugee children’s PA, only a few studies have focused on refugee children. This article presents an exploratory study on the spatial characteristics of six refugee accommodations in Berlin and their relation to school-aged refugee children’s opportunities to engage in PA. Micro-environmental attributes included building typology and availability, size, and access to communal PA spaces using Space Syntax. PA opportunities were assessed using staff surveys, interviews, and field trips. Results indicated that none of the case studies provided a comprehensive range of PA opportunities. They also revealed unequal access within the facilities. Whereas the role of size was inconsistent, vital predictors included fewer floors and corridors with easy access to internal and external PA spaces. Our recommendations include prioritizing compact buildings with moderate heights when retrofitting existing facilities and raising awareness for the importance of active play for this vulnerable group.

## 1. Introduction

Physical activity (PA) is a fundamental determinant of health and development for children. It helps build a robust body, stable mental health, and healthy relationships with peers [1,2,3,4]. Despite the strong evidence supporting the health benefits of PA and public health efforts to promote children’s PA, PA levels appear to be low among refugee children who have recognised refugee status or are asylum seekers [5]. A report showed that refugee children rarely met daily PA guidelines [6] such as engaging in 60 min of moderate to vigorous intensity PA per day [7]. Since refugee children live in unfamiliar and uncertain situations, being physically active can be highly beneficial [8]. It helps them build social ties with peers, transcend national boundaries, different cultures, and language barriers, and attain social inclusion [9,10,11]. Given the facts that refugee children’s numbers are increasing globally [12] and lack of PA can have long-term impacts on children’s health and development [13], it is critical to summarize empiric data and evidence-based recommendations that can promote PA among refugee children in this context. Increasing PA among this vulnerable group should be considered as a critical public health goal [14,15].

Multiple factors may be modified to facilitate children to be physically active. One relevant domain is the built environment, which refers to human-made spaces and structures in which people live, work/study, and engage in recreation on a day-to-day basis [16]. Several literature reviews have focused on micro (home) environmental attributes associated with non-refugee children’s PA, such as availability/access to exercise equipment [17,18,19,20]. However, the existing findings of environmental attributes relevant to non-refugee children’s PA may not apply to refugee children, as they live in very different settings.

As one of the top five hosting countries for refugees worldwide, Germany has accepted a total number 1.2 million refugees [21]; 6.5% of the asylum seekers arrive in Berlin, and more than one-third are minors [22]. Refugee families and their children are typically assigned to refugee camps or other temporary accommodations once they arrive in Germany [23]. Such facilities are often built in isolated and inaccessible areas of cities [24]. Even those granted long-term/permanent visas tend to have limited options for where to live, and they are more likely to reside in disadvantaged areas [6]. Furthermore, a report showed that those refugee children often spend a considerable amount of time inside refugee accommodations [6]. Studies also indicated that refugee children lack indoor space and communal facilities within the housing [25,26,27] for playing, or experience a lack of ‘dedicated play spaces’ inside the camp [28]. Due to such environmental arrangements, it is possible to argue that refugee children live in less favorable conditions for engaging in PA than non-refugee children; thus, it is possible that the built environment around refugee children hinders them from being active [14].

As mentioned above, since refugee children live in very different environmental settings compared to those experienced by non-refugees, to facilitate refugee children’s PA, it is necessary to further identify micro environmental factors in relevant contexts associated with their PA. The authors’ previous review has identified only a limited number of available studies (7 qualitative and 1 quantitative) which report that indoor and outdoor spaces in microenvironments are relevant to refugee children’s PA [14]. Within this frame of reference, this paper presents an exploratory study on spatial characteristics and PA opportunities of school-aged refugee children in six multi-type Berlin-located refugee accommodations. Based on these rare empiric data and insights, the first design recommendations will be established.

## 2. Social and Theoretical Framework

### 2.1. Refugee Accommodation Types and Asylsystem and in Germany

Figure 1 illustrates accommodation types for refugee families in the *Asylsystem*: refugees are designated to live in arriving centers or nearest available accommodations after application submission. The local state will then distribute them to an initial reception (Erstaufnahmeeinrichtung, German) as their first station in Germany [29]. Families are expected to stay here for no more than six months. Even so, most families stay longer. After application evaluations, most families will be settled in community accommodation (Gemeinschaftsunterkünften, German) for one to two years. Residential containers such as “Tempohomes” are built in Berlin to cope with inadequate living situations. They stay for a relatively unstable period until regular accommodations are available [30]. Private residences are possible after leaving initial receptions (e.g., in Berlin) or a specific evaluation, and differ between states [31]. Even today, *Asylsystem* is still updating, and new forms/prototypes of refugee accommodations are emerging.

### 2.2. External and Internal PA Space in Microenvironments

Research rarely focused on refugee accommodations as individual built environment levels [32,33]; several researchers argued that environmental scales are often ignored in debates over refugee studies, but specific urban spaces are essential for refugees navigating experiences of displacement and resettlement [34,35,36]. Therefore, a more detailed definition of environments should be developed. Bronfenbrenner’s ecological systems theory [37,38] has been applied as an environmental framework to understand refugee children’s day-to-day activities [39,40]. The built environments around refugee children include three environmental layers of interest: *microenvironment*; *mesoenvironment*, and *macroenvironment*. This study focuses on the microenvironment as the immediate vicinity of the child’s accommodation and contains the structures they directly contact in their daily lives as a home/refugee camp and its designated spaces [40]. Moreover, investigated PA spaces in microenvironments are divided into internal PA spaces as indicated playing spaces inside refugee accommodations (e.g., playroom, Figure 2a) and external PA spaces, which refer to outside designated playing spaces with this accommodation (e.g., playground, Figure 2b).

### 2.3. PA Space in Refugee Accommodations for Children and Space Syntax

The potential of employing space syntax as a spatial measurement tool related to refugee children’s PA seems multi-fold. It has the potential to extend the above-mentioned ‘refugee accommodations as microenvironments’ and reveal the spatial organization of facilities and their traffic flow, mutual intersections, and designated PA spaces. It could respond to the need to collect data on children residents and intrinsically should express how a configurational approach could influence children’s social and physical behavior. Space syntax can analyze the physical environments, calculate the relationships between spatial elements, and understand children’s behaviors (e.g., PA), relying solely on the spatial characteristics of built environments [41,42]. Previous research also showed that the potentiality of space syntax works with refugee camps’ built environments [43,44].

Furthermore, configurational approaches investigated by space syntax can objectively substantiate the development and improvement strategies for refugee accommodations. In this sense, space syntax can act as a framework for overcoming accountability, economic, and political constraints in such challenging contexts.

Application of space syntax in children’s research concerning playing remains novel; in a short opinion essay, Cutumisu and Spence [45] delineate why space syntax would be relevant in research relating to children’s PA. First, aspects of the environment such as a sense of place may influence children’s play, and space syntax has the capability to explain the walkability of the spatial configuration. Second, topology–geometric descriptors of space syntax resonate with how children develop and navigate spatial knowledge. The representation of built environments provided by space syntax might be fairer to children’s perceptions. In fact, the authors are aware of only one study that has applied space syntax in the context of children’s PA to associations between playground accessibility and playing [46]. The utilization of space syntax in understanding refugee children’s PA is relatively unexplored. This research is thereby contributing to the space syntax literature by exploring the broader application potential of its theories.

## 3. Methods

### 3.1. Sampling Strategy and Procedures

The sampling strategy of refugee accommodations is based on purposeful criteria [47]: available data, number of children residents (6 to 12 years old), types, sizes, and locations. The authors sent interview requests to 23 children-included refugee accommodations in Berlin; a detailed list was recorded, but could not be listed here due to the Berlin Data Protection Act [48], International Refugee Law [49], and protection terms of each accommodation for children. Six accommodations completed the data collection questionnaire, semi-structured interviews, and field trips, including three initial receptions, two community accommodations, and one Tempohome (see Figure 1), from July 2018 to February 2019 (Appendix A). The study sites were anonymized and named A to F based on interview times and accommodation types.

As shown in Figure 3, the research design is a mixed method consisting of (1) sampling and recruitment; (2) opensource data collection; (3) staff surveys (home manager and children staff) with field trips; (4) results and space syntax analysis; (5) comparative analysis; and finally (6) discussion and (7) conclusion. The process serves to validate software results through questionnaires and interviews in terms of analysis and its effects [50].

#### 3.1.1. Questionnaire

Face-to-face questionnaires were conducted by an interviewer asking questions of a respondent in person to allow the interviewer to explain and explore questions [51]. Participants may be more willing to give extended periods of their time in a face-to-face situation rather than via phone. They also have the opportunity to clarify wording and request more information for complex questions [52]. There were three blocks of questionnaires for home managers, including demographic information, basic situations, and the existing microenvironments. Questionnaires for children’s departments include basic situations, the existing microenvironments, refugee children’s daily lives, and a detailed daily PA timeline table. Additionally, staff were required to rate existing built environments for children’s PA with five-degree questions from 1 (worse) to 5 (excellent). All interviewees were fully informed about the research process (e.g., emails) and signed consent.

#### 3.1.2. Semi-Structured Interviews and Field Trips

As an effectively practical way of collecting environmental data with children-related staff [53,54], field trips were conducted to observe internal and external PA spaces, how activities happened in the accommodations, and PA programs provided by accommodations after questionnaires. Interviews were conducted while walking in the refugee accommodation for around 30 min. Questions were flexible and raised from questionnaires and observation, such as, “When is the playing room available for children?”. Six accommodations (excluding one, (E)) completed this process.

### 3.2. Measures of Children’s PA

This part was grounded in principle from the local authority or accommodation that refugee children’s playing needs to be under supervision of adults (e.g., partners, staff, volunteers). Precisely, the author investigated PA as ‘opportunities of PA for children’, including mainly two aspects by staff reports: (1) organized activity (e.g., play workshop, sports program) and (2) free play under supervision. A specific condition that was not covered above will be mentioned later.

### 3.3. Application of Space Syntax and Accommodation Spaces

Several different spatial measures, which investigated internal and external (indoor and outdoor) spaces, were selected for the analysis as below with SYNTACTIC [55] and DepthmapX [56] with floor plans for Section 4.1.

#### 3.3.1. Connectivity

Connectivity is a dynamic local measurement that measures the number of spaces immediately connecting to the space of origin [57]. To simplify, connectivity is the number of connected neighbors to the investigated space. It describes the relative level of control (in this context, accessibility of PA spaces) over the connected components [58]. Connectivity is chosen in this study because it is one of the most-used local measures and has been applied to refugee accommodation previously [42,43].

#### 3.3.2. Step Depth to Internal and External PA Spaces

Step depth illustrates which spaces are deeper and shallower than other spaces, related to the transitions formed by doors [57]. This measure has been used more often when investigating a specific space [59,60]. It was chosen for the analysis because it gives graspable descriptions when comparing spaces. For example, suppose the step depth of a living unit is six; readers know it takes six math steps from here to investigate PA spaces.

#### 3.3.3. Global Integration

Global integration describes the average depth of space to all other spaces in the system [57]. The spaces in a system can be ranked from the most integrated to the most segregated [61]. The calculation and investigation of spaces’ external and internal integration values in space syntax analysis help explain the occupant behaviour (such as PA) in the spatial model’s planning and organization [62].

## 4. Results and Comparison

Table 1 illustrates that A, B, and C are initial receptions, all addressed in existing buildings, and newly built accommodations are temporary containers, such as one Tempohome D and one community accommodation F. E was a former retirement home operated as a community accommodation. All accommodations had 18 to 30 school-aged refugee children. They were from various countries of origin, mainly Muslim countries. Moreover, four accommodations had already been closed when summarising this script (May 2022).

### 4.1. Spatial Characteristics

The internal PA space in accommodation A is one playroom (64 m^2^) on level 2 (Figure 4b,c), which had been repurposed from a former two-room living unit (Figure 4a). Children need to find the closest elevator/stairs and then get to level 2 for indoor playing. The playground is on level 0, with a total area of 390 m^2^. After passing through the canteen (only when it opens at mealtime), another corridor, or outside the main entrance, the children can arrive at this external PA space.

In accommodation B, there are two internal PA spaces on level 0. One is a playroom (1, 362 m^2^) for children near the main entrance (Figure 4b,c) and the other is a flexible open playing space (2, 65 m^2^) which is multi-functional (e.g., rock climbing or jumping from mat to mat). The living unit for a four-person family is 50 m^2^ (Figure 4a). By going through the nearest elevator or stair, children in building b go to level 0 for internal PA space or outside for external PA space. Children in building a need to go to another building for internal PA space. Three external PA spaces are linked together; 3 in Figure 4b,c is a functional playground with one ping-pong table, playable wood bridge, and sand playground (673 m^2^); for the rest of the playfields, 4 (665 m^2^) is next to the functional playground. As shown in Figure 4b, 5 is a rectangle square in an area of 937 m^2^.

Accommodation C represents a similar layout as A, where the main corridor runs in the middle and connects all living units on both sides. Children find the nearest elevator/stair, then go down to level 0 to reach internal or external PA spaces. The internal PA space is a playroom (41 m^2^) and links directly to a playground as an external PA space (Figure 4b,c). This playground (302 m^2^) provides various play equipment, such as a small ball playground, two ping-pong tables, and a sand playground with a slide. Children share it with other neighborhoods. A typical living unit is shown in Figure 4a of 42 m^2^. 

As shown in Figure 4b, layout representation is straightforward in this one-floor accommodation D, where living units are the centers and corridors run around them. Three small playgrounds around the containers are external PA spaces (1, 135 m^2^; 2, 148 m^2^; 3, 103 m^2^) filled with playable equipment. Children can easily reach outside playgrounds (photographs could not be taken due to the operator’s protection terms), but there is no internal PA space. As shown in Figure 4a, every living unit combination has a bathroom and a small kitchen with an area of 40 m^2^.

The layout is represented as a ‘tuning fork’ in four-floor accommodation E, where living units are as leaves and connected directly to branch-shaped corridors (Figure 4b). Internal PA space is housed in a separate building, with an area of 144 m^2^. The external playground (273 m^2^) is next to the internal PA space. A typical four-bed room for a family is 52 m^2^ with a balcony. Photographs could not be taken due to the operator’s protection terms of this accommodation.

In accommodation F, internal PA space is a four-container combination on level 0 of building b (Figure 4a, 52 m^2^), and a girl-specific playroom is under construction. There are various external PA spaces, such as a sand playground (465 m^2^, Figure 4b), a non-rectangular football field (2545 m^2^), a regular playground with PA equipment such as a slide, playing ring, and castle (720 m^2^, Figure 4c), and a basketball playground (420 m^2^). Typology here represents a ‘multi-tracks’ typology, similar to accommodation D, where the main corridor runs in the middle of the buildings (a, b, c). Children go outside accessing main corridors and go down through outdoor stairs on either side for outdoor playing. A typical living unit for a family is shown in Figure 4a, with a total area of 26 m^2^.

### 4.2. Opportunities for PA

Table 2 illustrates opportunities for PA in each accommodation. Children in A have the least PA opportunities, i.e., 1.5 h. Children in C have the second-least PA opportunities, around 2 h per workday. It is worth mentioning that since children in accommodation C share external PA space with their neighbors, play happens in turns; they may have a limited chance for immediate outdoor PA when other children in the neighborhood are already occupying the space. Opportunities for PA in accommodation E are around 2.5 h, either in internal or external PA spaces under supervision. Children in B and D (only in playground) have the second-most PA opportunities, at 4 h. Opportunities for PA in accommodation F are the most, around 5 h. Moreover, they can choose an organized activity between the playroom and sports in playgrounds every Friday.

### 4.3. Spatial Measure Analysis

The separate corridor is inconvenient to access PA spaces in accommodation A (average connectivity = 2.3) on each floor, making it difficult for children to reach external PA spaces (8.6) and internal PA spaces (10.1). Such spatial characteristics give low accessibility for PA spaces internally and externally.

Accommodation B is well-connected to internal (5.5) and external (4.8) PA space on average. Even though accommodation B is low in connectivity (average connectivity = 2.7), corridors are connected to living units. Given the size of the layout and the step depth to external and internal PA spaces, accommodation B provides good accessibility of internal and external PA spaces in microenvironments with identifiable, accessible, and prominent corridors.

Accommodation C has complex spatial characteristics resulting in low accessibility from living units to internal and external (10.4) PA spaces and average connectivity (2.2). Due to the unconnected corridors and multi-floors, it is difficult to reach internal and external PA spaces. Additionally, the shared external PA space with neighbors may reduce children’s PA.

Accommodation D consists of one-floor-containers, which can connect directly with external PA spaces. Such a spatial characteristic provides high PA space accessibility, especially multi-external PA space with an average step depth of 1.1. Significant parts of the living units are well-connected, with average connectivity of 4.5. However, there is no internal PA space in this accommodation, which may reduce the chances of PA when the situation (e.g., weather) is not suitable for outdoor playing.

Accommodation E has average PA space accessibility, with an average step depth of 5.3 and connectivity of 2.0. Even though the corridor is accessible, prominent, and well-integrated, separated internal and external PA spaces reduce accessibility for children to enter.

Accommodation F has good accessibility to external and internal PA spaces, with an average step depth of 3.5 and 5.0, respectively. The corridor on each floor of every building is integrated. It is possible for children from every building floor to efficiently and equally reach external PA space. Moreover, the external PA space is 2.0 or 3.0 step depths from the internal PA space.

### 4.4. Comparative Analysis

#### 4.4.1. Comparison between Opportunities for PA and Internal and External PA Spaces’ Size

Figure 5 shows differences between PA opportunities and PA space size for different accommodations. Children in accommodation F reported the most opportunities (5 h of PA), and F also had the most significant external PA space, with four different outdoor playgrounds and the fourth-biggest internal PA space. Accommodations B and D reported the second-most opportunities for PA at around 4 h per day. As mentioned previously, B had the most significant internal PA space, which opens every workday from 14:00 to 16:00. The second-biggest external PA space belonged to Accommodation D; however, it had no internal PA space. Children from accommodation C usually had 3 h of PA opportunities. They owned the second-smallest internal and external PA space, and various activities happened in the playroom every workday. Children in accommodation E spent 2 h per day on PA from 14:00 to 16:00 in their separate playroom and playground, and they could also attend organized activities every Saturday. Children in accommodation A reported the least time for PA opportunities (1.5 h).

In summary, internal and external PA space plays an essential role in refugee children’s opportunities for PA in microenvironments, serving as spaces for play. Differences and similarities were found in children’s PA concerning PA space size; however, there were no consistent results across all study sites by available cases.

#### 4.4.2. Comparison between Opportunities for PA and Spatial Measure Analysis

Figure 6 compares average step depth PA space, pointing to some similarities with typology, and Figure 7 compares spatial measure analysis to provide an overview of spatial characteristics. Both 11th-floor accommodation A and 10th-floor accommodation C had complex spatial layouts as separated corridors. They had similar spatial measure patterns, their step depth to the entire PA space was the highest, and integration values were the lowest among all sites. As shown in Figure 6, such a spatial characteristic gives low accessibility (higher step depths, Table 3, Figure 6) to PA spaces. It is also evident from these two refugee accommodations that floors are negatively related to children’s accessibility to PA spaces, and children here also have the shortest opportunities for PA.

Fifth-floor accommodations B and E had similar spatial layouts as sizable connectivity corridors that connected living units directly; the difference was that E had a separated building as the internal PA space. B was the only accommodation with spatial balance characteristics, such as multi-internal PA spaces and three large outside playgrounds (see Figure 5). Children’s opportunities for PA were 4 h here, and there was free play under supervision every workday. The opportunities for PA were 3 h per day in accommodation E, and organized sport happened every Saturday.

One-floor accommodation D had the highest connectivity and integration value, with multi-tracks typology. It was also nearest to the external PA space in step depth (see Table 3, Figure 6), since it had only one floor and most living units connected directly to external PA spaces. The disadvantage of this accommodation was that there was no available area for any indoor playing. Children had the second-most opportunities for PA, despite availability being limited to outdoor playing.

Three-floors accommodation F had the lowest step depth to internal PA space due to its clear layout (Table 3, Figure 6), and the corridor served as a breach that connected all living units with multi-tracks typology. Children had the most opportunities for PA at about 5 h per day. It also had the most considerable external PA space of the four playgrounds (see Figure 5). However, it had the smallest living unit, which was impossible for indoor playing.

#### 4.4.3. Comparison between Opportunities for PA and Staff Survey Ratings

Figure 8 shows that in accommodations with higher PA environment staff ratings, refugee children had more opportunities for PA. There were no consistent results by available cases. Moreover, it is necessary to look at the ratings and depth of the troughs formed by several individual answers:

Most staff (A, C, F) reported that internal PA spaces were regular living units; with limited area sizes, children did not have enough place for indoor playing. Staff from all accommodations (except D with no internal PA space) reported that they have an age distribution for the playroom time range (e.g., morning time is for preschool children, afternoon is for school-aged, and night time is for teenagers). Additionally, staff from B and F mentioned that they had gender-specific PA and specific girls’ playing time; for example, there was a ‘girl’s night’ every week in accommodation F since girls tend not to appear in the mix-gender playroom as a culture/religion consideration. Thus, the precise mechanisms of how the variables work require further investigation; however, what matters regarding built environments for refugee children’s PA seems to be the levels of open and accessible corridors. More precisely, the accessibility for refugee children from their living units to external and internal PA spaces is most important. It could be considered that higher accessibility provides refugee children with more chances for active playing.

Studying the existing microenvironments for refugee children’s PA is a highly complex analysis. Due to limited samples, it is difficult to construct typologies to analyze spatial characteristics. However, this empirical material and study will contribute to refugee children’s research field, specifically for existing built environments concerning their PA.

## 5. Discussion

### 5.1. Access to Internal and External PA Spaces

Official policy documents indicated the importance of having playing areas and showed the ambition to plan sufficient playing spaces in refugee accommodations [8,63]. However, in the six case studies, existing conflicts between planning and realism emerge. Examining the internal PA spaces, most accommodations had similar problems: playing rooms inside these accommodations were retrofitted from regular living units and had a limited area size (A, C, F) and there was one accommodation with no internal PA space (D). Most external PA spaces are simply fenced open spaces with limited playing equipment (B, E). It is possible to discern a trend where internal PA spaces seemingly have been placed some distance away from most living units, which was also exposed by the official report [6]. Potentially, this may be done in order to avoid noise disturbance. It is an important insight that this positioning also results in PA spaces being less accessible. Policymakers/designers should be aware of such conflicting interests, including the importance of accessible playing spaces in reality.

Research rarely investigated the internal spatial characteristics of refugee accommodations concerning their activities [42]. Only a few studies shared that playing areas inside refugee accommodations are related to children’s playing [25,26,27,28]. A deeper understanding of internal and external PA spaces and children’s PA can be confirmed from the quantitative analyses within this study; it can be seen as the first set of circumstantial evidence regarding play spaces and their role in microenvironments. It is thus suggested that a potential continuation of this study disserts qualitative interviews with refugee children and their parents to better understand their views on spaces for play [15] (pp. 105–122). Another possible extension of this study would be to assess accessibility differences, including more samples and refugee accommodation types, such as Modular accommodation for refugees [64], to better understand how accessibility varies among different accommodation types.

### 5.2. Space Syntax and Its Representation

This study applied highly-summarized ways of the metric distance to the internal and external PA spaces (see Table 3 and Figure 7). It is based on comparing scale diagrams over different case studies and introducing this professional method to a broader target audience. However, it should be emphasized that this analysis will change with variations of corridors and degrees of fencing, not forgetting the realities of the analysis assumes a flat site. Consequently, the conclusions drawn are from a typical typology; specific conditions in refugee accommodations (e.g., reconstruction) were not covered and have not been considered in this study.

### 5.3. Strengths and Limitations of the Research

There were several limitations due to the explorative nature of this study. Firstly, the authors have had limited ability to gain access to the appropriate type or geographic scope of participants. It was a rather tricky task to approach refugee accommodations with children in Berlin: less than 1/3 responded to interview requests, and most of the responding accommodations refused participation without giving any reason. This research had a small sample size; therefore, a limitation for the volume of data collection within the scope of this study. Thus, the study site was not random sampling, but involved refugee accommodation participants willing to collaborate in this research. This may raise the issue of identifying significant relationships in the data and whether these cases were truly representative. However, this study aimed to provide insights into the relationship between spatial characteristics and refugee children’s PA. Hence, what is generalizable from this study is more of an analysis than the direct results. In general, this is a cross-sectional study, and depending on the scope, relevant studies on this topic are limited; however, in this case, this study can also be considered a significant opportunity to identify literature gaps and present the need for further development in this research area. A longitudinal or experimental study should be developed in the future to confirm the relationship between environmental factors and refugee children’s PA. One possibility is to track PA before and after transitions to refugee accommodation as a quasi-experimental study.

#### 5.3.1. Issues for PA Measures

The research had a subjective measurement for PA. It is evident that self-report measures contain errors and bias in capturing physical activity [65]; the ‘opportunities for PA’ have limited the ability to conduct a thorough analysis of the results: it only represented the overall time range refugee children spent on PA but could not be sub-divided to PA types. Objective measurement design is unsuitable and could not be achieved in the current social context due to legal, privacy, and refugee-specific terms and the research country. Future studies should benefit from existing studies targeting non-refugee children, as they have developed various methods to assess PA [66]. Additionally, since there was limited access to refugee children and their parents, the author had to apply surveys with staff from children’s departments instead. In this context, the author assumed that the children’s staff were experts in refugee children’s daily lives whom they were supervising. However, most of the investigated refugee accommodations faced frequent staff turnover situations, and some of the staff surveys might be finished by several interviewees.

#### 5.3.2. Issues for Environmental Measures

The study did not examine the quality aspects of play areas. For example, the presence of play equipment, lighting, and maintenance can be related to the use of spaces. The defined spatial measures had the potential to work as a forward predictor of built environments for refugee children’s PA; however, this is still an early stage problem and there is not enough material from research fields to justify or support parts of the researcher’s views.

## 6. Implications for Research and Practice

### 6.1. Future Research Directions

This research field requires more quantitative studies to understand better environmental features conducive to refugee children’s PA. Future research studies should consider in-depth data collection on a large environmental scale (e.g., meso), more quantitative studies with PA measures, and larger sample sizes. Below are specific research themes that deserve detailed investigations.

Studies with refugee accommodations with similar spatial characteristics (e.g., former use) and sizes but different building shapes may help to better understand the spatial typology and the relations of PA spaces’ sizes;Studies should employ objectively derived (e.g., wearable sensor) measures or validated self-report measures of relevant PA. Particular attention could focus on specific attributes of PA levels (sedentary, moderately, vigorously);Spatial quality measures (e.g., PA equipment types; lighting) should be investigated and evaluated. A measurement (e.g., mathematical calculation) should be established with more empirical material and available research in microenvironments;Conduct longitudinal studies which track refugee children’s PA lives when they relocate from a temporary refugee facility to long-term accommodations;Future research should relate spatial layouts typology to investigate whether these spatial characteristics could influence refugee children’s daily PA in detail or individuals.

### 6.2. Recommendations for Refugee Accommodations Operators

The refugee accommodation operators should be a stable force between the objective and local authorities, contributing to the gap between existing refugee accommodations’ spatial limitations and potentialities. The staff expressed their specific concerns and worries about refugee children’s PA in the interviews; advice summarized from research and empirical materials should be passed to the operators:Refugee accommodations should provide better playrooms and playgrounds, and the size, form, and positioning of these places should be carefully considered;Refugee accommodations could provide more ‘easy to reach’ internal and external PA spaces, for instance, multi-locations; operators can indicate the importance of accessible playgrounds and playrooms from the beginning of proposals;Concerns about noise disturbance for nearby living units are relevant, but can be addressed by further measures, including opening hours and noise insulation;For those refugee accommodations with limited resources for organized activity, free playing under supervision should be considered a regular daily schedule for refugee children to extend the opportunities for PA for refugee children.

### 6.3. Recommendations for Architects and Urban Planners

Internal PA space is vital for refugee children’s PA. These designs should be considered in detail and meet the flexibility and multi-options of refugee children’s PA (e.g., gender-specific playroom);Urban planners and architects should suggest that existing buildings with easy access to PA spaces, i.e., clear, open, and accessible corridors with fewer floors, should be prioritized for retrofitting to refugee accommodations. High-rise buildings have been identified in this research as a barrier to refugee children’s active playing and thus should be carefully considered in refugee accommodation choices;It is suggested that open and straightforward spatial characteristics could contribute to built environments for refugee children’s PA. Furthermore, the highlighted spatial characteristics associated with refugee children’s PA could be considered as potential measures during a new refugee accommodation’s design process.

### 6.4. Recommendations for Refugee Policy Makers

Local agencies, particularly those with a coordinating role in area regeneration, need to incorporate methods for securing refugee children’s participation in their everyday practice. It should include the spectrum of participatory activities from seeking and providing information to full engagement in the more formal structures of the organizations, using methods that promote inclusion. Ultimately, the participation of refugee children should be regarded as a prerequisite by policymakers for ensuring high-quality policy decisions and delivery;Microenvironments for refugee children’s PA should be considered a primary measure when reconstructing refugee accommodations set in existing buildings. For those newly built refugee accommodations, a microenvironment that supports refugee children’s PA should be considered a benchmark in the design process.

## 7. Conclusions

This study provided new evidence and insights into spatial characteristics in microenvironments and refugee children’s PA. Moreover, it accurately described and depicted the role of spatial characteristics in shaping their opportunities to engage in daily PA. The analysis can benchmark design strategies for spatial characteristics in microenvironments to evaluate PA environments of existing buildings’ potentialities as refugee accommodations.

This analysis has used a freely available software within the established Space Syntax framework that allows related practitioners to assess the likelihood of evaluating PA environments using floor plan or scale map inputs. This analysis will make it possible to lead informed discussions among related practitioners about the impact of their design solutions and hopefully give evidence-based designs a new direction.

Understanding spatial characteristics of refugee accommodations associated with refugee children’s PA was crucial, as it highlighted the spatial properties of built environments for refugee children’s PA and indicated that spatial characteristics could also have a direct or indirect effect. Policymakers have multiple choices for refugee accommodations’ construction/setting. Architects showed their favor of specific refugee accommodations typologies based on multi-reasons; however, their effect on built environments for refugee children’s PA was under-researched before. This study clarifies the relationship that could be used to assess evaluation schemes better. With a better understanding of this topic, related practitioners now have more empirical material and detailed directions to further investigate social–spatial innovation and regeneration to create more healthy and inclusive cities. These efforts are essential in minimizing the current health and spatial inequality issues these vulnerable groups face across Germany and worldwide.

## Figures and Tables

**Figure 1 ijerph-19-07756-f001:**
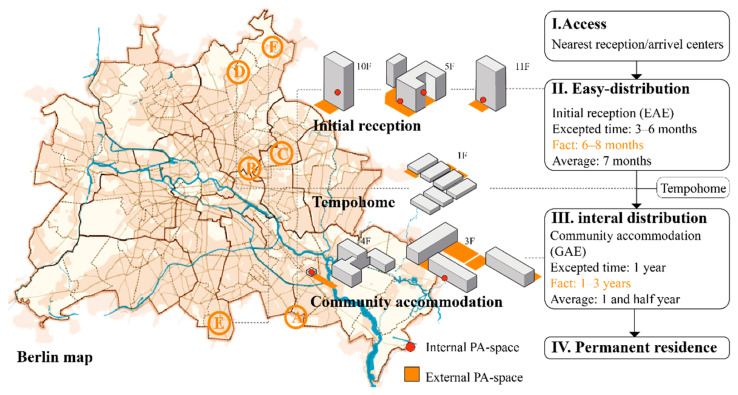
Location and type of refugee accommodations featured in the six case studies (**left**) as part of the different phases of the current *Asylsystem* in Germany (**right**). Source: United Nations Children’s Fund, Federal Office for Migration and Refugees, and State Office for Refugee Affairs Berlin reports.

**Figure 2 ijerph-19-07756-f002:**
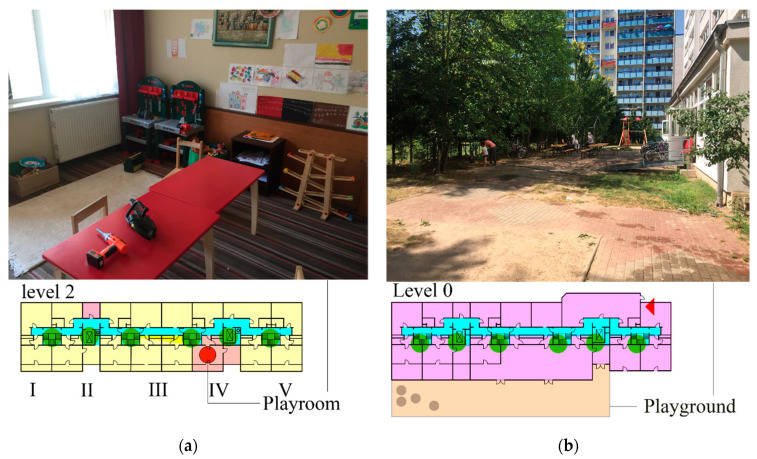
Two examples of PA spaces in accommodation A: (**a**) internal PA space; (**b**) external PA space.

**Figure 3 ijerph-19-07756-f003:**
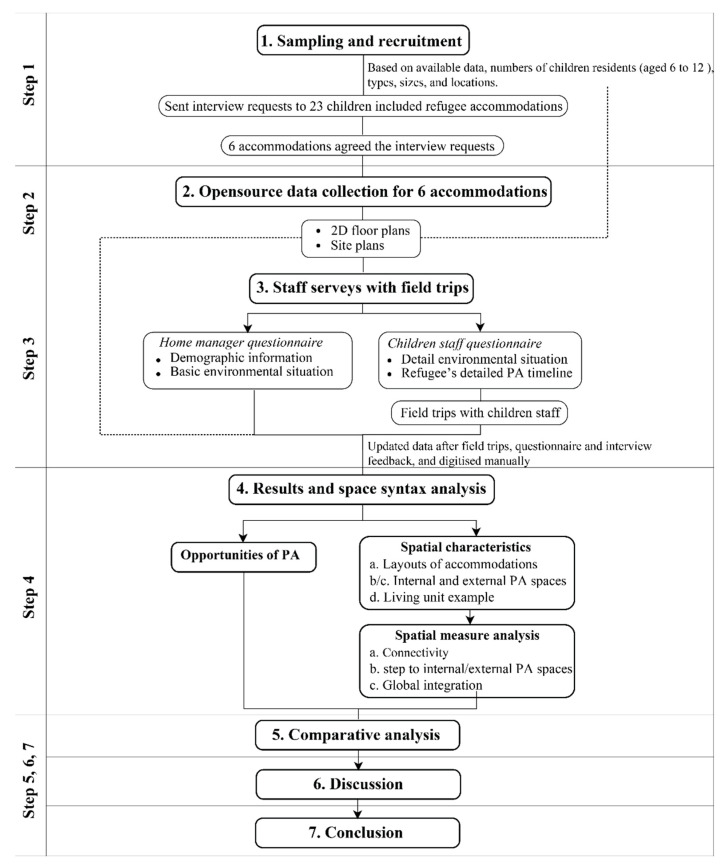
Overview of microenvironments approach including data sources and outputs.

**Figure 4 ijerph-19-07756-f004:**
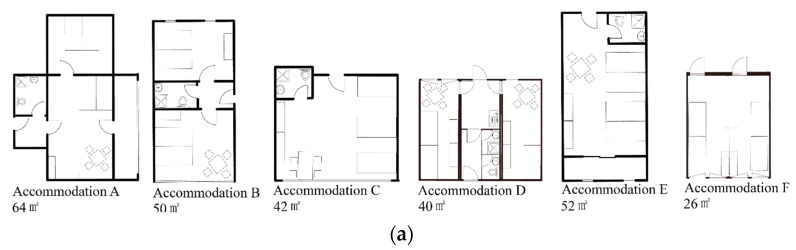
(**a**) living unit example of each accommodation; (**b**) layouts in the microenvironment; (**c**) internal PA space and external PA space photos (except accommodation E).

**Figure 5 ijerph-19-07756-f005:**
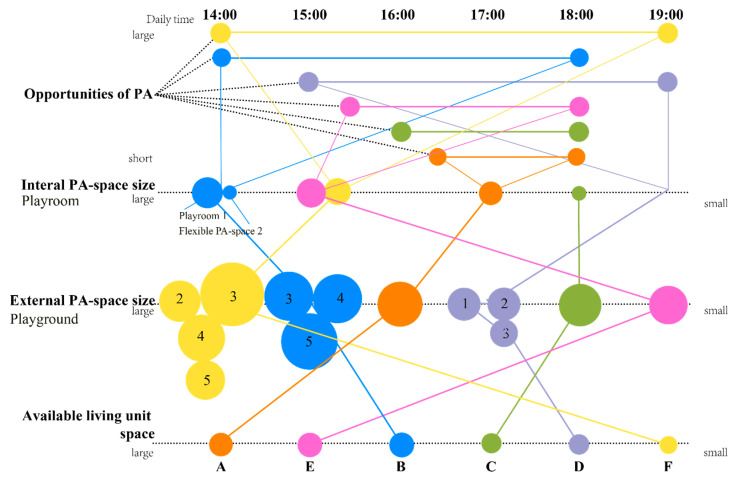
Comparison between opportunities for PA and PA space size of the six study sites.

**Figure 6 ijerph-19-07756-f006:**
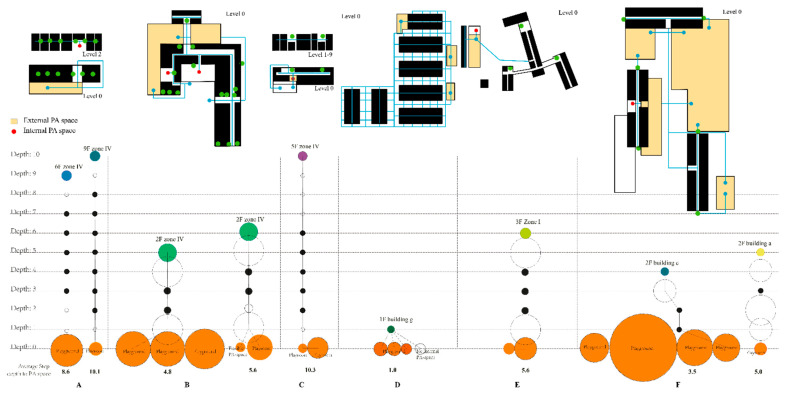
Average step depth example from living unit to internal and external PA space and typology.

**Figure 7 ijerph-19-07756-f007:**
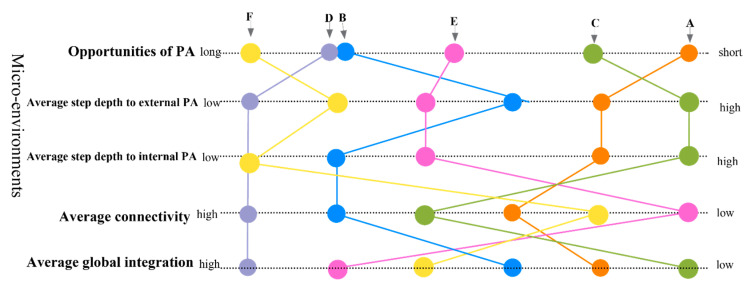
Comparison of overall findings in spatial measures of six study sites.

**Figure 8 ijerph-19-07756-f008:**
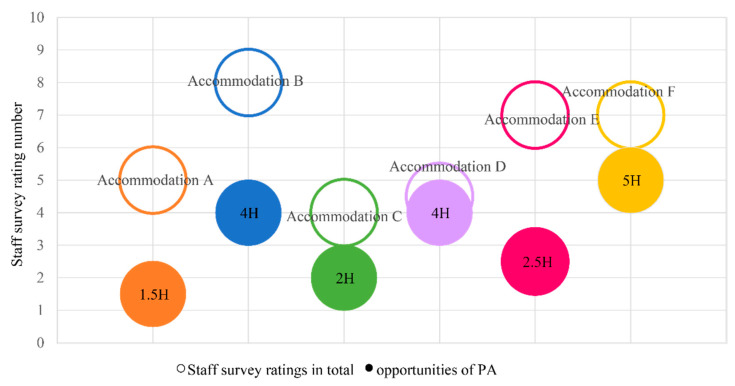
Patterns of staff survey ratings and opportunities for PA.

**Table 1 ijerph-19-07756-t001:** Comparison of accommodation type, former use, floor, existing period, children population, and countries of origin across 6 case studies.

Accommodation	Type	Former Use	Floor	Population ^1^|Capacity|Living Unit Numbers	Existing Period	Children Population: Aged 6–12 ^1^	Children’s Countries of Origin ^1^
A	initial reception	Hotel	11	250|400|100	December 2015–current ^2^	30	Muslim and Asia
B	initial reception	Health care facility	3	490|500|170	September 2014–August 2019	33	Asia and Africa
C	initial reception	Residential block	10	217|350|127	February 2012–current	27	Multiethnic
D	Tempohomes	Newly built containers	1	170|256|64	December 2016–July 2019	20–30	Muslim
E	community accommodation	Retirement home	4	200|265|90	July 2015–October 2020	18	Multiethnic
F	community accommodation	Newly built containers	3	424|560|251	April 2015–September 2020	30	Muslim

^1^ By each individual interview time; ^2^ current: May 2022; there may be operator subject alter during different periods (The accommodation may be still open, but runs by different operator compares to interview time).

**Table 2 ijerph-19-07756-t002:** Opportunities for PA: types and categories.

Accommodation	Opportunities for PA (Staff Report)	Monday	Tuesday	Wednesday	Thursday	Friday	Saturday
A	16:30–18:00	○ 16:30–18:00, playroom	○ 16:30–18:00, playroom	○ 16:30–18:00, playroom • Biweekly, football, playground	○ 16:30–18:00, playroom	○ 16:30–18:00, playroom	
B	16:00–18:00	○ 16:00–18:00 in playroom					
C	16:00–18:00	○ 16:00–18:00, playroom	14:00–18:00, playroom	○ 16:00–18:00, playroom	○ 15:00–18:00, playroom	○ 14:00–18:00, walking around (meso) together	
D	15:00–18:00			○ 17:00–19:00, football, playground			○ 17:00–19:00, football, playground
E	15:30–18:00						○ 14:00–16:00, football/badminton/jump rope, playground
F	15:00–19:00	○ 14:00–16:00, playroom	○ 14:00–16:00, playroom		○ 14:00–16:00, playroom	○ 14:00–16:00, playroom • 14:00–16:00, family football, playground	

•, organised activity; ○, free play under supervision.

**Table 3 ijerph-19-07756-t003:** Spatial measure values: average connectivity, step depth to internal/external PA spaces, and integration of each case study A to E.

Average Connectivity	Average Step Depth		Average Integration
	Step Depth to External PA-Space	Step Depth to Internal PA-Space	
2.3	8.6	10.1	0.8
2.7	5.5	4.8	0.9
2.2	10.4		0.7
4.5	1.1	no internal PA-space	3.1
2.0	5.3		1.4
2.1	5.1	3.5	1.1

## Data Availability

Not applicable.

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
