# Peer review of "Understanding Spatial Characteristics of Refugee Accommodations Associated with Refugee Children’s Physical Activity in Microenvironments: Six Case Studies in Berlin"

_ijerph, 2022, doi:10.3390/ijerph19137756_

Round 1

Reviewer 1 Report

Dear,

The authors bring a contribution through investigation about spatial characteristics in environments and physical activities to refugee children. I believe that on Introduction section a strong review with more details bring a real tehrotical support for tha paper . I observed some mistakes, related to the abstract. Is a necessary better explanation about the objectives and results.

The authors cited the first paragraph “PA” initials. And on the second paragraph tha authors explain the initials

The context on physical activity of children approached in the present paper is a totally compreenshive and relevant practical and academic contribution. However, it is not clear what the objective proposes.  

“Evidence and insights into spatial  characteristics and school-aged refugee children’s PA abstract from six multi-type Berlin located refugee accommodations”: Please re-written the sentence.

Why the Figure 3 was called before than figure 2? Please fix theorder of the figures

3 Methods section: please, detail better the procedure to interview made with 23 children

The Figure 2 doesn't explain which the steps are presented

In general speaking, the Methodology steps must be amplified. There is little information related to the procedure used by the authors.

Table 1: please describe the propose this information contained on table. The authors cited as “comparison demographic”, but not bring more details.

Reviewer 2 Report

Line 134-148

Questionnaire and field trips are appropriate and efficient research method for conducting staff surveys and evaluating PA spaces in this study which may need further interpretations.

Line 178

It is clear that 'Results and comparison' shows the analysis and details of individual accommodation. More discussion and comparison based on the accommodations with same type and same former use could be added in order to deliver more idea and future directions for improving PA spaces in the accommodations with same type and same former use.

Line 397-398

As the suggestions provided in this study (interviews with refugee children and their parents to understand their views on spaces for play better), is it possible to provide a pilot interview of the children and their parents (if there is any feedback from them) which may offer in-depth feelings and experience from the residents.

Round 2

Reviewer 1 Report

Dear atuhors,

The Introduction needs some careful review and still is shortly. Basically, three paragraphs were inserted on body text in this section.
In the Methods section improvements were made.
